# The Effect of Denosumab on Rotator Cuff Repair in Women Aged 60 and over with Osteoporosis: A Prospective Observational Study

**DOI:** 10.3390/biomedicines12051069

**Published:** 2024-05-12

**Authors:** Ki-Tae Kim, Sanghyeon Lee, Ho-Won Lee, Shi-Hyun Kim, Yong-Beom Lee

**Affiliations:** 1Department of Orthopedic Surgery, Hallym University Sacred Heart Hospital, College of Medicine, Hallym University, Anyang 14068, Republic of Korea; kimkt8399@naver.com (K.-T.K.); bklp00@gmail.com (S.-H.K.); 2Department of Orthopedic Surgery, Kangnam Sacred Heart Hospital, Medical Center, Hallym University, 1 Singil-ro, Yeongdeungpo-gu, Seoul 07441, Republic of Korea; osgardener@naver.com (S.L.); lhwghm@gmail.com (H.-W.L.)

**Keywords:** rotator cuff tear, re-tear, osteoporosis, shoulder arthroscopy, bone marrow density

## Abstract

Background: In previous studies, denosumab, a RANKL human monoclonal antibody used in osteoporosis treatment, has shown efficacy in tendon healing after rotator cuff repair. This prospective study investigated the effects of denosumab on tendon healing, re-tear rates, and clinical outcomes post rotator cuff repair in women with osteoporosis. Method: This was a prospective, observational study, employing propensity score matching for the control group. From March 2018 to March 2023, female patients over the age of 60 with normal bone density undergoing arthroscopic rotator cuff repair were selected as controls through propensity score matching (PSM) and compared with female patients of the same age group with osteoporosis who were receiving denosumab treatment. The control group was matched using 1-to-2 propensity score matching. Radiological examinations and functional outcomes were assessed preoperatively and at 6 months postoperatively. Results: In the final analysis, the study comprised 34 patients in the denosumab treatment group (Group 1) and 68 patients in the control group (Group 2). The functional scores showed significant improvement at 6 months post-surgery in both groups. No significant difference in the functional scores was observed among the groups. The re-tear rate, defined according to Sugaya’s classification (types IV and V) as re-tear, was slightly higher in Group 1 at 16.7% (6 of 34) compared to Group 2 at 11.7% (8 of 68), but the difference was not statistically significant (*p* = 0.469). The re-tear patterns, classified according to Rhee’s classification, also showed no significant difference among the groups (Group 1: 2/4 of 6; Group 2: 4/4 of 8; *p* = 0.571). The occurrence of type I re-tear exhibited no significant difference between the two groups (5.9% vs. 5.9%; *p* = 1.000). Conclusions: The administration of denosumab following arthroscopic rotator cuff repair in women aged 60 and over with osteoporosis resulted in a re-tear rate that was similar to that observed in patients without osteoporosis. This result suggests that denosumab administration might be beneficial for rotator cuff healing, particularly in the context of osteoporosis, a known risk factor for increased retear rates. Therefore, comprehensive osteoporosis screening and treatment should be considered in conjunction with rotator cuff repair surgery in middle-aged women.

## 1. Introduction

Rotator cuff tears are a significant concern within the field of orthopedics, leading to decreased shoulder function and pain, thus diminishing the quality of life for many individuals. The natural healing capability of rotator cuff tears is known to be limited. With the advent of advanced imaging diagnostics, such as MRI and ultrasound, along with improvements in surgical techniques and arthroscopic tools, the frequency of performing rotator cuff repair surgeries has been on the rise. Consequently, the incidence of re-tears is also common, with reported rates varying between 10 and 91% depending on the size and type of the tear [1,2,3].

Among the various factors associated with the re-tearing of rotator cuff repairs, osteoporosis has been identified as a significant risk factor [4,5,6]. In patients with osteoporosis, the overall quality of bone is compromised, including at the greater tuberosity, and the rotator cuff attachment site, leading to complications with suture anchors and a decrease in tendon-to-bone healing due to increased bone resorption [7].

Recent studies have addressed these issues by exploring the effect of osteoporosis treatment on the re-tear rates following rotator cuff repair surgery [8,9,10,11,12,13,14]. Junjie Xu et al. have shown that in a rotator cuff tear rat model, administering abaloparatide and denosumab during the repair surgery resulted in significant improvements in the failure load and stiffness of the rotator cuff tendon–bone interface eight weeks postinjury. They also observed increased new bone formation, bone mineralization, and expression of genes related to mineralized tissue. Furthermore, we previously reported that treating osteoporosis in patients with rotator cuff tears using zoledronate can reduce re-tear rates and that teriparatide treatment can enhance tendon-to-bone healing in the recovery process of rotator cuff tear patients [15].

Denosumab, a powerful osteoporosis drug, acts as a monoclonal human antibody targeting the Receptor Activator of Nuclear Factor-Kappa B Ligand (RANKL), thereby inhibiting bone resorption [16]. Denosumab is recommended as a first-line treatment for osteoporosis by the latest clinical guidelines from the American Association of Clinical Endocrinology/American College of Endocrinology and the Endocrine Society (AACE/ACE). It is frequently used because of biannual subcutaneous administration that improves patient compliance [17,18].

This research investigated the impact of denosumab, administered subcutaneously after surgery, on the success of rotator cuff repairs, specifically its ability to lower re-tear rates and improve both functional outcomes and clinical scores. Although animal studies suggest denosumab aids in tendon–bone healing, clinical evidence in humans remains scarce. We hypothesized that the use of denosumab in women over the age of 60 diagnosed with osteoporosis undergoing Arthroscopic Rotator Cuff Repair (ARCR) would not only aid in tendon–bone healing but also contribute positively to clinical outcomes, including reducing the re-tear rate. We conducted a prospective, observational study to compare outcomes in patients undergoing surgery with those receiving denosumab postoperatively. This study is the first investigation into denosumab’s role in preventing re-tears following rotator cuff repair surgery.

## 2. Materials and Methods

### 2.1. Study Design

This study was conducted as a prospective observational study targeting elderly women with osteoporosis who were undergoing repair for full-thickness rotator cuff tears with postoperative administration of denosumab. Institutional review board approval was obtained from Hallym University Sacred Heart Hospital (HALLYM2019-04-017-001), with all participants providing informed consent prior to their inclusion. From March 2018 to March 2023, participants who met the following inclusion criteria were enrolled: (1) women aged 60 years and above; (2) diagnosis of full-thickness rotator cuff tears confirmed by oblique-coronal preoperative MRI; (3) diagnosis of osteoporosis, as defined by a Bone Mineral Density (BMD) score equal to −2.5.

The exclusion criteria for the trial included the following: (1) lack of MRI evaluations pre- and postsurgery, or contraindications to MRI; (2) follow-up loss within six months after surgery; (3) history of previous surgery on the affected side of the arm or shoulder; (4) additional repair techniques other than transosseous equivalent repair due to irreparability, including superior capsular reconstruction, patch augmentation, and partial repair; (5) diagnoses of infections, autoimmune diseases, systemic skeletal conditions, rotator cuff arthropathy, severe osteoarthritis, or rheumatoid arthritis; (6) prior use of specific osteoporosis treatments, excluding vitamin D or calcium; (7) contraindications to osteoporosis medications due to conditions such as hypersensitivity, hypocalcemia, pregnancy, or severe kidney dysfunction (creatinine clearance < 35 mL/min).

To evaluate the therapeutic outcomes, the treatment group was compared with a control group comprising patients without osteoporosis who, consequently, did not receive denosumab. Apart from BMD scores, the selection was based on patients treated for rotator cuff injuries at our facility between March 2019 and March 2023 who matched the study’s eligibility requirements. The main outcome of interest was the incidence of re-tears observed within a six-month period post-rotator cuff repair surgery. It was postulated that osteoporotic patients receiving denosumab might exhibit re-tear frequencies comparable to those in nonosteoporotic patients. Secondary outcomes were also reviewed, including evaluations of clinical efficacy and shoulder function.

### 2.2. Surgical Approach and Denosumab Administration

All operations were performed by a surgeon who has over 20 years of experience in the field. The sizes of the rotator cuff tears were evaluated using an arthroscopic probe. In instances of dislocation or tear of the biceps tendon’s long head, tenotomy or tenodesis was performed. Acromioplasty was also uniformly applied to excise acromial spurs. The rotator cuff footprint was prepared with a motorized burr to remove soft tissue remnants and facilitate bone–tendon healing. Medial row anchors (Y-knot RC, ConMed, Utica, NY, USA) were inserted along the medial edge of the footprint near the articular surface, and lateral anchors were set according to the bridge’s design, utilizing various knotless anchors for lateral fixation.

In the treatment arm, patients received a 60 mg subcutaneous injection of denosumab (AMGEN, Seoul, Republic of Korea) two days following the surgery, administered either in the upper arm of the unoperated side or in the abdomen. Patients were monitored for side effects such as fever, myalgia, arthralgia, and headache, and oral acetaminophen was provided for symptom management as needed.

A structured rehabilitation program was initiated post-surgery, beginning with pendulum exercises one week after the procedure. Active-assisted Range of Motion (ROM) exercises started six weeks postoperation, transitioning to isometric muscle strengthening exercises using a rubber band at three months. Gradual reintroduction to minor sports activities was advised between three and six months post-surgery, contingent on the recovery of full ROM and adequate muscle strength, tailored to the individual’s tear size and recovery rate.

### 2.3. Outcome Measurement and Data Collection

To determine the presence of rotator cuff re-tears, MRI examinations were conducted prior to surgery and again at the six-month postoperative mark. These MRI scans were performed using a 3.0-Tesla scanner (Signa HDx; GE Healthcare, Milwaukee, WI, USA) equipped with a specialized shoulder coil. It was noted that the initial preoperative MRI scans for some patients were obtained at external facilities before their initial consultation at our clinic. During scanning, patients were positioned supine with their forearm in a neutral orientation. The MRI protocol included axial fast spin-echo proton-density images with fat suppression, oblique coronal fast spin-echo T2-weighted images with fat suppression, and sagittal fast spin-echo T2-weighted images with fat suppression, with specific parameters tailored to optimize image quality for each sequence.

The integrity of the repaired tendon was assessed using the Sugaya classification system based on T2-weighted oblique coronal MRI images, ranging from type I (indicating robust, uniformly low-intensity cuff thickness) to type V (indicating a significant discontinuity suggestive of a medium or large tear) [19]. A re-tear was defined as findings consistent with Sugaya type IV or V. Furthermore, the pattern of re-tears was categorized using Rhee’s classification system on T2-weighted oblique sagittal scans, distinguishing between completely detached cuff tissue (type I) and partially attached remnant cuff tissue at the insertion site (type II) [20]. The extent of fatty degeneration within the torn muscle was quantified using Goutallier’s five-stage grading system on T1-weighted MRI images, considering its impact on repair success rates [21]. Measurements of tear size and location were precisely recorded, given their relevance to the likelihood of surgical failure. All MRI analyses were conducted by two independent reviewers, with any discrepancies resolved through consensus discussion. The Global Fatty Degeneration Index (GFDI) was calculated as the average of the Goutallier grades for the supraspinatus, infraspinatus, and subscapularis muscles, providing a comprehensive assessment of muscle degeneration [22].

Bone Mineral Density (BMD) screenings were performed for all eligible patients using dual-energy X-ray absorptiometry (DXA) technology (Lunar Prodigy Advance, Encore, GE Medical System, Milwaukee, WI, USA), focusing on the lumbar spine (L1–L4) and the proximal femur but omitting the femur’s ward area.

To evaluate the functional outcomes of patients, we utilized a suite of assessments including the Simple Shoulder Test (SST), the University of California, Los Angeles (UCLA) Shoulder Rating Scale, the American Shoulder and Elbow Surgeon (ASES) score, Constant–Murley Shoulder Score (CSS), and measurements of shoulder flexibility [23,24,25,26]. Functional assessments were conducted by a dedicated research assistant at baseline and again six months postprocedure. Shoulder flexibility, encompassing forward flexion and the degree of internal rotation, was quantified with a goniometer. The benchmark for internal rotation was identified as the highest spinal level the thumb could comfortably reach, with vertebrae numerically labeled from the sacrum’s base (0) up to the fourth thoracic vertebra. The same assistant consistently performed these measurements to ensure reliability.

### 2.4. Statistical Analysis

To minimize the impact of potential confounders, we utilized Propensity Score Matching (PSM), ensuring comparable clinicopathological features across both the treatment and control groups. The study’s required sample size was determined based on prior research findings. Notably, Chung et al. observed a marked difference in re-tear rates between osteoporotic patients with a BMD below −2.5 (41.7%) and those with a BMD above −2.5 (13.2%; *p* < 0.001). With the expectation that the treatment group’s re-tear rate would be comparable to the control group, a treatment-to-control ratio of 1:2 was selected. After adjusting for a 10% drop-out rate, we established a target sample size of 21 for the treatment group to achieve 90% power and an error of 0.05.

A 1:2 propensity score matching using the K-nearest neighbor method was employed, incorporating several parameters: patient age, gender, extent of physical activity (defined as engaging in exercise more than two days per week), and muscle involvement (including individual or combinations of the supraspinatus, subscapularis, and infraspinatus). Variables also included the size of the tendon tear determined through MRI, preoperative fatty degeneration assessed separately for the supraspinatus, infraspinatus, and subscapularis muscles, and the comprehensive Global Fatty Degeneration Index. Additional factors considered were the patient’s smoking history (specifically if they smoked regularly within the past year) and any documented instances of trauma.

All statistical analyses were performed using SPSS version 26 (IBM Corp. Released 2023. IBM SPSS Statistics for Windows, Version 29.0.2.0 Armonk, NY, USA: IBM Corp Functional assessments were analyzed with the Student’s *t*-test and paired *t*-test for comparisons among groups and comparisons in the same group using numerical data, respectively. χ^2^ and Fisher’s exact tests were applied for the categorical variables. To compare the re-tear rate among the groups, we used the Noninferiority (NI) test with a noninferiority margin of 10% according to half of the upper margin of the confidence interval in Oh’s study. Statistical significance was set at *p* < 0.05.

## 3. Results

### 3.1. Patients

In the prospective trial, 42 patients who satisfied the inclusion criteria were enrolled in the study. During the surgical procedures, patients needing further interventions such as patch augmentation (five patients), superior capsular reconstruction (one patient), and revision surgery on the same shoulder (one patient) were excluded from the study. Additionally, one patient was lost to follow-up. As a result, the data analysis included 34 patients who received denosumab treatment (Group 1).

For the control group, there were 358 elderly female patients who underwent rotator cuff repair surgery. Patients requiring patch augmentation (27 individuals), superior capsular reconstruction (15 individuals), or revision surgery (22 individuals), as well as those diagnosed with osteoporosis or osteopenia (60 individuals), were excluded from the study. Propensity score matching was subsequently conducted for the remaining 187 patients to form the control group. Using a one-to-two matching strategy, 68 patients were allocated to Group 2. The patient selection algorithm utilized in this study is illustrated in Figure 1.

After the PSM, demographic analysis showed no significant differences among the groups, except for the BMD T-score, which was notably lower in Group 1 (−2.83 ± 0.75) compared to Group 2 (−0.97 ± 1.20; *p* < 0.001). The extent of preoperative fatty degeneration in both groups was comparable for the supraspinatus (*p* = 0.226), infraspinatus (*p* = 0.334), and subscapularis muscles (*p* = 0.762). The Global Fatty Degeneration Index also showed no significant variance, being 1.27 ± 1.01 for Group 1 and 1.05 ± 0.96 for Group 2 (*p* = 0.193). Furthermore, the preoperative tear size was slightly larger in Group 1 (28.13 ± 9.6) than in Group 2 (26.30 ± 10.9), though this difference did not reach statistical significance (*p* = 0.426) (Table 1).

### 3.2. Re-Tear Rates and Clinical Outcomes

Significant enhancements in functional scores, including the SST, UCLA shoulder score, ASES, and CSS, were observed in both groups from baseline to six months post-surgery, with these improvements achieving statistical significance. Preoperatively, except for the UCLA score, clinical scores were generally lower in Group 1, but by the six-month postoperative assessment, all scores—aside from the ASES score, including the SST, UCLA, and CSS scores—trended higher in Group 2, albeit without significant differences. Significant gains were seen in the Range of Motion (ROM) metrics, such as forward flexion, internal rotation, and external rotation. Prior to and following surgery, forward flexion was marginally higher in Group 2, and internal rotation was slightly greater in Group 1, though these variations lacked statistical significance. The occurrence of re-tears stood at 11.7% (6 out of 34 patients) in Group 1 and 8.8% (8 out of 68 patients) in Group 2, with the confidence interval at −0.09 to 0.218 using a Noninferiority (NI) analysis, signifying a meaningful outcome beyond an NI margin of 10%. The distribution of re-tear patterns, categorized as type I in two instances and type II in four instances for Group 1 (versus four cases of type I and four cases of type II in Group 2), did not display statistical significance (Table 2).

## 4. Discussion

Our study results suggest that although the administration of denosumab did not make a discernible difference in clinical outcomes, the lack of a significant difference in the re-tear rate implies it could be beneficial in mitigating the risk of re-tears linked to osteoporosis, a factor identified as a risk in previous research.

To date, there is a paucity of research on the effect of treating concomitant osteoporosis on rotator cuff recovery following surgical repair [14]. Although animal studies have suggested that denosumab may aid in tendon–bone interface healing at the site of rotator cuff repair, the clinical efficacy postsurgery remains largely unreported, leaving its effectiveness subject to debate. This study represents the first human subject research to examine the impact of denosumab on the rotator cuff repair site, incorporating a wide array of variables, such as demographic factors, tear patterns, tear sizes, and fatty infiltration (patient-dependent factors), alongside surgical repair techniques (surgeon-dependent factors). The consistency in surgical procedures was ensured by having all surgeries performed by a single surgeon, maintaining uniformity in the type and positioning of the anchors, as well as their angle positions across the majority of procedures. To control for the myriad factors influencing rotator cuff re-tear, 1–2 propensity scoring matching was employed, facilitating a nonrandomized prospective study design. Our findings indicate no significant difference in the re-tear rates between patients receiving denosumab treatment and those without osteoporosis, thus contributing valuable insight into the ongoing discussion regarding the role of osteoporosis treatment in the context of rotator cuff repair.

Research into rotator cuff re-tear following surgical repair remains a hotly debated topic within the field of orthopedics. Studies on the risk factors for rotator cuff re-tear have been widely reported through various regression analyses and meta-analyses [27,28,29,30,31]. In the majority of these studies, osteoporosis is identified as a primary risk factor for re-tear, as well as for failures in rotator cuff repair surgery due to anchor loosening and pull-out [4]. Hong et al. reported in a population-based cohort study that the risk of rotator cuff tear was 1.79 times higher in patients with osteoporosis [6]. Similarly, Kim et al. found a significantly higher re-tear rate in patients with lower Bone Mineral Density (BMD), suggesting that BMD is an independent factor affecting rotator cuff healing [31]. In cadaveric studies, Tingart et al. indicated that the tensile strength of anchors is closely related to the quality of bone [32]. Furthermore, Lee et al. demonstrated a significant association between the BMD of the Greater Tuberosity (GT), as well as tear size, and the risk of cutting through. In our study cohort, no instances of anchor failure were observed [33]. This outcome can be attributed to the effect of denosumab, which inhibits the activity of osteoclasts, thereby facilitating the integration of the anchor and promoting the healing of the tendon–bone interface. By enhancing stiffness, denosumab may reduce the tensile forces acting on the anchor, potentially lowering the risk of anchor failure. Our findings suggest that addressing the underlying osteoporosis with denosumab could be a crucial strategy in improving the outcomes of rotator cuff repair surgeries by mitigating key risk factors for re-tear and anchor-related complications.

Denosumab is an antibody-targeting RANKL, playing a crucial role in inhibiting osteoclast formation and activation by intervening in the RANKL–RANK signaling pathway [34,35]. Osteoclast precursors express RANK, the receptor for RANKL. When RANK binds with RANKL, expressed either by osteoblasts or osteoclasts, it induces the differentiation of osteoclast precursors into mature osteoclasts [36]. Osteoprotegerin (OPG), a decoy receptor produced mainly by osteoblasts, acts on the RANKL–RANK signaling pathway to inhibit the formation and bone resorption activity of osteoclasts. This understanding has positioned the inhibition of RANKL–RANK signaling as a primary target in the treatment of osteoporosis. Utilizing this mechanism, denosumab, a human monoclonal IgG2 antibody against RANKL, was developed. Unlike IgG1 antibodies, IgG2 does not induce complement-dependent cytotoxicity or antibody-dependent cellular cytotoxicity. Additionally, denosumab exhibits specificity and high affinity for RANKL without interacting with other members of the Tumor Necrosis Factor (TNF) ligand family [37]. The efficacy of Prolia (i.e., denosumab) has been demonstrated in large-scale clinical trials, showing increased bone density and fracture prevention effects at various sites, including the spine, non-spine, and hip regions. The FREEDOM (Fracture Reduction Evaluation of Denosumab in Osteoporosis Every Six Months) study and its extension have reported the effectiveness and safety of denosumab [38]. Common side effects of denosumab include joint pain, muscle pain, infections such as soft tissue inflammation, hypocalcemia, osteonecrosis of the jaw, and an increased risk of atypical femoral fractures. However, in our study cohort, none of the patients treated with denosumab experienced these side effects.

Various studies have explored the impact of osteoporosis treatments on the healing process following rotator cuff repair surgery. Kim et al. reported that raloxifene enhances rotator cuff healing in animal models [8]. Similarly, animal experiments conducted by Junjie Xu and colleagues assessed the biomechanical properties of the rotator cuff repair site in osteoporotic rats, comparing abaloparatide and denosumab to a control group. Both treatments were found to promote rotator cuff healing, with abaloparatide showing superior outcomes compared to denosumab [14]. Human studies, such as the one by Oh et al., have indicated that the administration of teriparatide (a recombinant human parathyroid hormone) significantly reduces the re-tear rate after rotator cuff surgery, suggesting that parathyroid hormone derivatives, which promote bone formation, could further enhance the biomechanical healing of the tendon–bone interface [39]. However, because of denosumab’s inability to recognize rodent RANK-L, alternative methods are used in experiments, making it challenging to predict whether similar outcomes would occur in humans. Research involving bisphosphonates has shown their potential to accelerate rotator cuff healing in animal studies, yet a large-group retrospective study by Cancienne, J. M. et al. involving 1343 patients reported no significant difference in the revision rate between treated patients and those with osteoporosis who did not receive treatment. This included a range of bisphosphonate medications [12,40]. In our research team’s study, we specifically looked at zoledronate and found no significant difference in the re-tear rate between the treatment group and a control group without osteoporosis, suggesting an ongoing debate over their effectiveness. This study is the first human subject research comparing re-tear rates post rotator cuff repair in patients treated with denosumab to those without osteoporosis, finding no significant difference in outcomes [15].

Among patients undergoing rotator cuff repair surgery, many who receive concurrent osteoporosis treatment are women over the age of 60. In Korea, because of insurance criteria, bisphosphonates and denosumab are more frequently used as first-line treatments over PTH or romosozumab. Given their widespread use globally, this research can serve as a valuable guide for clinicians when considering osteoporosis treatment in patients undergoing rotator cuff repair surgery, indicating that managing osteoporosis may not influence the re-tear rate post-surgery.

Research on agents that promote healing of the rotator cuff repair site continues to be reported. There is ongoing debate about the impact of vitamin D on rotator cuff healing. Chen et al. have reported a significant correlation between vitamin D deficiency, higher re-tear rates, and early postoperative pain [41]. However, Ryu et al. found no association between lower serum vitamin D levels and rotator cuff healing and poor outcomes, whereas Oh. et al.’s comparison of serum and tissue vitamin D levels indicated that vitamin D is associated with postoperative muscle power and reduced fatty degeneration but not with surgical failure [10,42]. In our study, all patients in both the treatment and control groups were administered 100 mg of calcium and 1000 IU of vitamin D daily; hence, the effect of these variables could not be isolated for assessment. Additionally, there have been reports suggesting the efficacy of estrogen and testosterone in rotator cuff healing, but patients receiving hormone therapy were excluded from our study to control for these effects, and serum levels were not measured separately, rendering their impact indeterminable.

The rate of re-tear following rotator cuff repair surgery is reported in the literature to range from 11% to 94%, with a recent meta-analysis indicating an overall mean re-tear rate of 21% (ranging from 7% to 37%) at 6–12 months follow-up. In our study, at a minimum of 6 months follow-up, the denosumab user group showed a re-tear rate of 16.7%, whereas the control group had a rate of 11.7%, aligning with the re-tear rates reported in existing studies. Rhee and colleagues reported that the risk of re-tear increases with tear sizes larger than 2 cm because of increased tendon tension [43]. Murrell et al. analyzed the association between age and rotator cuff re-tear, reporting re-tear rates of 10% for ages 50–59, 15% for ages 60–69, and a sharp increase to over 25% for ages 70 and above [44]. Le et al. identified tear size as the primary factor contributing to re-tear through multivariate regression analysis, with age being the next significant factor [27]. In our study, the treatment group had an average age of 68.35 ± 7.27 and tear size of 28.13 ± 9.6 mm, whereas the control group had an average age of 65.37 ± 8.35 and tear size of 26.30 ± 10.95 mm, with the majority being patients over 65 years old and with medium to large tears of more than 2 cm, showing similar re-tear rates to those reported in the literature.

This study was conducted exclusively on postmenopausal women aged around 60 years and above. The National Health Insurance Service in Korea prohibits the prescription of Bone Mineral Density (BMD) testing for men under 70 years old, except in cases where pathological fractures are suspected. Furthermore, most male patients diagnosed with osteoporosis tend to have a high incidence of comorbidities. To maintain the homogeneity of the group, these patients were excluded from the study. Gender and age were controlled to solely assess the effect of the osteoporosis treatment drugs.

## 5. Limitation

This study has several limitations. First, to definitively prove the effect of denosumab on rotator cuff re-tear, the highest level of evidence would come from comparing denosumab-administered groups with nonadministered groups in a randomized fashion among patients with osteoporosis. However, in Korea, all postmenopausal women over the age of 60 receive osteoporosis treatment through health insurance. Given this context and the ethical concerns of not treating diagnosed but untreated osteoporosis, we could only compare the denosumab-administered group with a nonadministered group not diagnosed with osteoporosis. Second, the small sample size presents limitations in proving the effectiveness of denosumab post-rotator cuff repair surgery. Third, there is a scarcity of control group numbers. To control for the age variable, female patients of similar age to the treatment group, even if not diagnosed with osteoporosis, mostly had accompanying osteopenia, necessitating a two-fold matching to designate a control group with as normal bone density as possible. Fourth, the follow-up period was very short at six months. This duration is not sufficiently long to evaluate the re-tear rate, functional assessment, or long-term safety of denosumab, and does not account for the effects of multiple administrations, necessitating further research with longer follow-up and MRI evaluations. Fifth, this study did not control for variables related to other systemic diseases. Therefore, future research could yield better results by reducing the selection bias through randomized controlled trials with larger samples.

## 6. Conclusions

In elderly women aged 60 and over with osteoporosis, the administration of denosumab following arthroscopic rotator cuff repair showed no difference in the failure rate of the repair compared to patients with normal bone density. This result suggests that denosumab administration might be beneficial for rotator cuff healing, particularly in the context of osteoporosis, a known risk factor for increased retear rates. This finding underscores the necessity of meticulous osteoporosis screening and proactive treatment in patients with rotator cuff tears. Such measures are vital to enhance the success of surgical interventions and promote better postoperative outcomes in this patient demographic.

## Figures and Tables

**Figure 1 biomedicines-12-01069-f001:**
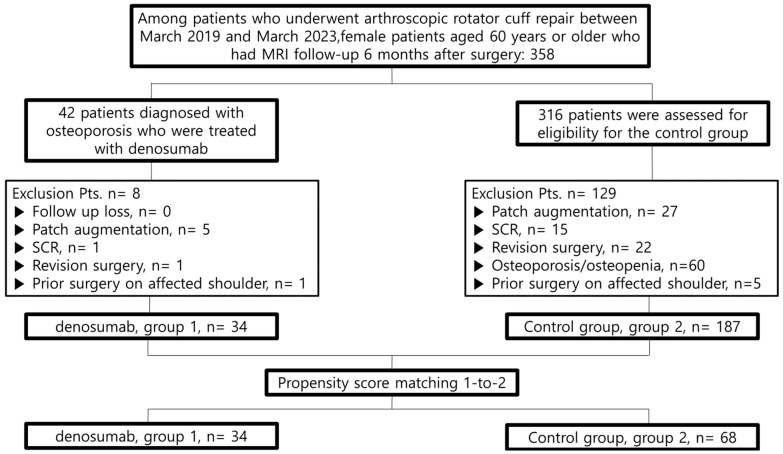
Flowchart of patient selection.

**Table 1 biomedicines-12-01069-t001:** Demographic data.

	Group 1	Group 2	*p*-Value
No. of patients	34	68	
Age, mean ± SD, years	68.35 ± 7.27	65.37 ± 8.35	0.07
Sex, male/female, *n*	0/34	0/68	
Onset, mean ± SD, mo.	9.57 ± 21.27	11.62 ± 15.67	0.582
Bone mineral density (T-score), mean ± SD	−2.83 ± 0.75	−0.97 ± 1.20	<0.001
Preoperative fatty degeneration, mean ± SD			
Supraspinatus	1.94 ± 1.34	1.59 ± 1.39	0.226
Infraspinatus	0.88 ± 0.80	0.69 ± 0.99	0.334
Subscapularis	1.01 ± 1.25	0.88 ± 1.08	0.271
Global Fatty Degeneration Index	1.27 ± 1.01	1.05 ± 0.96	0.193
Tear size, mean ± SD, mm	28.13 ± 9.69	26.30 ± 10.95	0.426
Smoking history, yes/no, *n*	34/0	67/1	0.667
Trauma history, yes/no, *n*	14/20	23/45	0.467
Regular exercise, yes/no, *n*	27/7	58/10	0.452

**Table 2 biomedicines-12-01069-t002:** Comparison of clinical and radiographic outcomes between Groups 1 and 2.

	Group 1	Group 2	*p*-Value
Clinical outcomes			
SST			
Preoperative	4.29 ± 2.57	5 ± 2.74	0.191
Postoperative at 6 months	6.67 ± 2.71	7.26 ± 2.27	0.298
*p*-Value	0.003	<0.001	
UCLA			
Preoperative	16.75 ± 7.57	16.3 ± 5.84	0.947
Postoperative at 6 months	24.17 ± 6.34	24.74 ± 7.28	0.724
*p*-Value	<0.001	<0.001	
ASES			
Preoperative	51.23 ± 17.02	52.75 ± 17.44	0.488
Postoperative at 6 months	73.22 ± 18.33	73.31 ± 15.05	0.948
*p*-Value	<0.001	<0.001	
CSS			
Preoperative	40.17 ± 12.11	41.68 ± 16.10	0.33
Postoperative at 6 months	60.35 ± 17.13	63.25 ± 15.20	0.473
*p*-Value	<0.001	<0.001	
ROM			
Forward flexion			
Preoperative	132.31 ± 54.74	140.41 ± 53.97	0.232
Postoperative at 6 months	176.54 ± 5.61	174.69 ± 16.47	0.601
*p*-Value	<0.001	<0.001	
Internal rotation			
Preoperative	4.32 ± 4.75	3.83 ± 4.43	0.437
Postoperative at 6 months	9.32 ± 5.99	7.76 ± 6.16	0.28
*p*-Value	0.003	<0.001	
Radiologic outcomes			
Retear, % (*n*/N, 95%CI)	6/34, 16.7%	8/68 11.7%	0.469
Retear pattern, *n*, types I/II	2/4	4/4	0.571

## Data Availability

The data presented in this study are available on request from the corresponding author. The data are not publicly available due to privacy concerns related to patient data.

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
