# Peer review of "The Effect of Denosumab on Rotator Cuff Repair in Women Aged 60 and over with Osteoporosis: A Prospective Observational Study"

_biomedicines, 2024, doi:10.3390/biomedicines12051069_

Round 1

Reviewer 1 Report

Comments and Suggestions for Authors

The authors claimed that they studied the effect of denosumab (Den) on rotator cuff repair in women with osteoporosis (OP).

However, the study is incorrectly designed. The conclusions do not reflect the results. The authors should have a control group of OP patients not treated with denosumab. Otherwise, the conclusions do not reflect the results. If it is not possible to get a control group of OP patients without Den treatment, the authors could conclude that OP does not affect rotator cuff repair. However, this conclusion could be wrong. Another option, the authors should conduct additional study to examine how Den treatment affects rotator cuff repair during 6 month using examination of some clinical traits.

The present version of the study has no sense.

Author Response

Thank you for your insightful comments regarding our study design and the conclusions drawn. We acknowledge the critical points you have raised about the need for a control group of osteoporosis patients not treated with denosumab to more definitively assess the impact of Denosumab on rotator cuff repair.

In our study, the control group comprised women over the age of 60 who were not diagnosed with osteoporosis. This was highlighted in the first paragraph of the limitations section of our manuscript. Your suggestion is precise and underscores a crucial aspect of clinical research design that ideally would include such a control group to directly measure the effect of Denosumab on rotator cuff healing in osteoporosis patients.

Given the ethical and logistical constraints in not treating diagnosed osteoporosis in postmenopausal women, as outlined in our limitations, this was not feasible in our context. If osteoporosis indeed acts as a risk factor that impedes rotator cuff healing, our findings could indirectly suggest a potential beneficial effect of Denosumab in women with osteoporosis on rotator cuff healing.

We appreciate your recommendation for conducting additional studies to examine how Denosumab treatment affects rotator cuff repair over 6 months, focusing on specific clinical traits. This suggestion is valuable and will be considered for future research to further elucidate the relationship between Denosumab treatment and rotator cuff repair outcomes.

Again, thank you for your constructive critique, which will undoubtedly help in refining our research and approaches in future studies.

Reviewer 2 Report

Comments and Suggestions for Authors

The authors investigated whether denosumab, an anti-osteoporotic monoclonal antibody, could lower retear risk and enhance clinical-functional outcomes in postmenopausal women after rotator cuff repair surgery. Their non-randomized prospective study revealed that osteoporotic patients receiving a single postoperative denosumab dose achieved comparable results, both clinically and radiographically, to a non-osteoporotic control group not treated with denosumab. They advocate for vigilant osteoporosis management in cuff repair patients to mitigate retear risk. The study is well-structured, with the use of propensity score matching compensating for the lack of randomization. The methodology is comprehensive, results are well-documented, and the discussion aligns with the findings. However, it remains uncertain whether a single denosumab dose given postoperatively is sufficient, prompting consideration of alternative dosing strategies and timing to optimize outcomes. Specifically, the authors should discuss if multiple denosumab doses administered before surgery would yield different results.

Author Response

Thank you for your thorough review and constructive feedback on our study

Regarding your concern about the sufficiency of a single postoperative dose of denosumab, we acknowledge that this is an important aspect that merits further exploration. This is considered a limitation of the study, and it has been modified and attached in the main text from line 388 to line 391.

Your suggestion to investigate the effects of multiple doses of denosumab administered before surgery is highly valuable. Preoperative dosing could potentially influence the baseline bone quality and thus the surgical outcomes more significantly. We agree that this approach might provide insights into optimizing pre-surgical preparation for better tendon-bone healing and overall functional recovery.

In response to your recommendation, we plan to design a follow-up study that will compare different dosing schedules of denosumab, including multiple preoperative doses, to assess their impact on the outcomes of rotator cuff repair surgery. This study will aim to fill the gaps identified in our current research and provide a more comprehensive understanding of denosumab's role in managing osteoporosis in the context of orthopedic surgeries.

Thank you once again for your insightful feedback, which is instrumental in refining our research direction and methodologies for future studies.

Reviewer 3 Report

Comments and Suggestions for Authors

The Authors describe the clinical effects of denosumab on the treatment of rotator cuff tear. This molecule that is a monoclonal antibody acts as a potent anti-osteoporosis and in this study its clinical effect have been studied in old patients with rotator cuff tear. The manuscript scientifically sounds and the results appear convincing. This molecule seems to have a positive effect in tendon healing. Statistical analysis shows the efficacy of this drug. 

I think that this manuscript deserves to be published after two minor points.

i) in the literature, denosumab is described as drug potentially related to subcutaneous necrosis, which is a side effect. The authors should better describe this possibility and verify in their patients this occurrence. 

ii) tha administration of denosumab is in subcutaneous way, but the authors should be more precise in the point of administration. Have they tried the intrarticular injection? They could discuss this possibility.

Author Response

Thank you for your thoughtful review and positive evaluation of our manuscript. We appreciate your acknowledgment of the scientific soundness of our study and the compelling nature of our results.

Comment 1) Regarding your first point about the potential side effects associated with denosumab: We recognize the importance of discussing potential side effects when using pharmacological treatments. In our study, we closely monitored all patients for adverse effects following administration. Fortunately, no cases of necrosis were observed in our patient cohort. This has been noted in the Discussion, lines 304-307. We are grateful for your insightful comments.

Comment 2) Concerning the second point about the route of denosumab administration: In our research, denosumab was administered subcutaneously, in accordance with approved guidelines and existing practices for the treatment of osteoporosis. We have addressed your points in the manuscript (lines 121-125).

We did not explore intra-articular injections of denosumab in this study. However, your suggestion raises an interesting point about alternative routes of administration that could potentially deliver therapeutic effects more directly to the site of injury. Further research into the efficacy and safety of various denosumab delivery routes in this specific clinical application could lead to additional study designs.

Round 2

Reviewer 1 Report

Comments and Suggestions for Authors

Comments were not addressed. Reject.